# Administration of Enfortumab Vedotin after Immune-Checkpoint Inhibitor and the Prognosis in Japanese Metastatic Urothelial Carcinoma: A Large Database Study on Enfortumab Vedotin in Metastatic Urothelial Carcinoma

**DOI:** 10.3390/cancers15174227

**Published:** 2023-08-23

**Authors:** Takashi Kawahara, Akihito Hasizume, Koichi Uemura, Katsuya Yamaguchi, Hiroki Ito, Teppei Takeshima, Hisashi Hasumi, Jun-ichi Teranishi, Kimito Ousaka, Kazuhide Makiyama, Hiroji Uemura

**Affiliations:** 1Departments of Urology and Renal Transplantation, Yokohama City University Medical Center, 4-57 Urafune-cho, Minami-ku, Yokohama 232-0024, Japankatsuya333@gmail.com (K.Y.); teppei_t@yokohama-cu.ac.jp (T.T.); jteran@yokohama-cu.ac.jp (J.-i.T.); hu0428@yokohama-cu.ac.jp (H.U.); 2Department of Urology, Graduate School of Medicine, Yokohama City University, Yokohama 236-0027, Japan; uemura.koi.pf@yokohama-cu.ac.jp (K.U.); hiroki22@yokohama-cu.ac.jp (H.I.); hasumi@yokohama-cu.ac.jp (H.H.); makiya@yokohama-cu.ac.jp (K.M.)

**Keywords:** enfortumab vedotin, urothelial carcinoma, database, real world, immune checkpoint

## Abstract

**Simple Summary:**

Enfortumab vedotin, a targeted therapy for advanced urothelial carcinoma, has shown efficacy, especially in those treated with platinum-based chemotherapy and immune-checkpoint inhibitors. The EV-301 phase III trial reported enhanced overall survival and response rates than conventional chemotherapy. However, its effectiveness in Japanese patients needs further real-world validation. Analyzing 6007 urothelial cancer patients treated with pembrolizumab, 563 subsequently received enfortumab vedotin, while 443 switched to docetaxel or paclitaxel. Results indicated the enfortumab vedotin group had an extended overall survival compared to the paclitaxel/docetaxel group (*p* = 0.013, HR: 0.71). Conclusively, enfortumab vedotin offers Japanese patients better overall survival prospects after pembrolizumab treatment than docetaxel or paclitaxel.

**Abstract:**

Background: Enfortumab vedotin shows promise as a targeted therapy for advanced urothelial carcinoma, particularly in patients who have previously received platinum-based chemotherapy and an immune-checkpoint inhibitor. The EV-301 phase III trial demonstrated significantly improved overall survival and response rates compared to standard chemotherapy. However, more data, especially from larger real-world studies, are needed to further assess its effectiveness in Japanese patients. Methods: A total of 6007 urothelial cancer patients inducted with pembrolizumab as a second-line treatment were analyzed. Among them, 563 patients received enfortumab vedotin after pembrolizumab, while 443 patients received docetaxel or paclitaxel after pembrolizumab, and all were included in the study for efficacy as a life prolonging agent. Results: The enfortumab vedotin group showed a longer overall survival than the paclitaxel/docetaxel group (*p* = 0.013, HR: 0.71). In multivariate analysis, enfortumab vedotin induction was the independent risk factor for overall survival (*p* = 0.013, HR: 0.70). There were no significant differences in cancer-specific survival. Conclusions: Enfortumab vedotin prolonged the overall survival for Japanese advanced or metastatic urothelial carcinoma patients compared to paclitaxel or docetaxel after pembrolizumab treatment.

## 1. Introduction

Advanced urothelial carcinoma poses significant challenges in terms of diagnosis, treatment, and patient outcomes [1,2,3]. This aggressive form of urothelial carcinoma often presents at an advanced stage, leading to limited treatment options and a poor prognosis. In recent years, molecular biomarkers such as programmed death-ligand 1 (PD-L1) expression and fibroblast growth factor receptor (FGFR) alterations have gained importance in identifying potential targets for personalized therapies. Treatment options for advanced urothelial carcinoma have evolved significantly in recent years. Platinum-based chemotherapy, typically with cisplatin or carboplatin, has been the standard of care for eligible patients [4]. However, a significant proportion of patients may not be candidates for platinum-based chemotherapy due to factors such as renal impairment or poor performance status. Even when patients are eligible for platinum-based systemic chemotherapy, these advanced urothelial carcinomas are refractory to this chemotherapy. For these patients, immune-checkpoint inhibitors targeting PD-1/PD-L1 have shown efficacy. But, most cases are refractory to immune-checkpoint inhibitors [5]. In these patients, enfortumab vedotin, an antibody-drug conjugate targeting Nectin-4, is used, which has shown promising results in clinical trials and received regulatory approval as a subsequent therapy [6,7].

Enfortumab vedotin is an innovative and promising targeted therapy that has shown significant efficacy in the treatment of advanced urothelial carcinoma in adult patients who have previously received platinum-containing chemotherapy and a PD-1 or PD-L1 inhibitor. Building upon the encouraging results of EV-201, the EV-301 phase III trial compared enfortumab vedotin to standard chemotherapy (docetaxel, paclitaxel, and vinflunine) in patients with locally advanced or metastatic urothelial carcinoma who had previously received platinum-based chemotherapy and a PD-1/PD-L1 inhibitor [7,8,9]. The trial demonstrated a significantly improved overall survival (OS) with enfortumab vedotin compared to chemotherapy, establishing it as a superior treatment option. The median OS in the enfortumab vedotin group was 12.9 months compared to 9.0 months in the chemotherapy group (HR:0.70). Moreover, enfortumab vedotin showed a higher ORR of 40.6% versus 17.9% with chemotherapy.

Despite this effectiveness of enfortumab vedotin in urothelial carcinoma, the EV-301 study only included 86 Japanese patients (50 in the chemotherapy group and 36 in the enfortumab vedotin group) and there were no large real-world data in terms of enfortumab vedotin. This study investigated the effectiveness of enfortumab vedotin as a post-pembrolizumab treatment for advanced or metastatic Japanese urothelial carcinoma patients using a large medical insurance database [7,10].

## 2. Materials and Methods

This study used the database of healthcare fees, which covers around 25.6% of DPC hospitals in Japan [11]. These data were obtained from Medical Data Vision (Tokyo, Japan). From April 2008 to December 2022, we extracted the patients who were diagnosed with urothelial carcinoma by using ICD-10 codes. A total of around 239,685 urothelial cancer patients were extracted from this database, and 6007 were inducted with pembrolizumab. Regarding these patients, their age, cancer location, types of previous chemotherapy, types of post-chemotherapy, and prognosis were analyzed.

A total of 6007 patients were introduced to pembrolizumab as a second-line treatment. For analyzing the efficacy of enfortumab vedotin compared to paclitaxel or docetaxel, neither cases with pembrolizumab monotreatment nor cases who were treated with enfortumab vedotin, paclitaxel, or docetaxel before pembrolizumab were excluded in this study. Finally, 563 patients received enfortumab vedotin after pembrolizumab, and 443 patients, including 97 docetaxel and 346 paclitaxel patients, received docetaxel or paclitaxel after pembrolizumab and were all analyzed in this study.

### Statistical Analyses

The participants’ characteristics and scores were analyzed using Mann–Whitney U tests (PCL/DOC vs. EV cohort). The OS and cancer-specific survival (CSS) were determined using a Kaplan–Meier curve, and a log-rank test was used as a comparison. Multivariate analysis was used to compare the risk factors for OS and CSS. These tests were conducted using the Graph Pad Prism software program (Graph Pad Software, version 10, La Jolla, CA, USA). *p*-Values of <0.05 were considered to indicate statistical significance.

## 3. Results

A total of 1006 patients including 443 paclitaxel/docetaxel cases and 563 enfortumab vedotin cases were enrolled in this study (Figure 1). There were no gender and age differences observed in this study (male: 79.0% in paclitaxel/docetaxel and 77.1% in enfortumab vedotin *p* = 0.446, age (median (mean ± SD)): 73 (71.4 ± 9.9) in paclitaxel/docetaxel and 73 (72.2 ± 8.3) in enfortumab vedotin *p* = 0.187). There were also no differences in the location of the urothelial carcinoma. Overall, 279 (63.0%) cases in paclitaxel/docetaxel and 336 (59.7%) in enfortumab vedotin were seen in the bladder (*p* = 0.287); 98 (22.1%) in paclitaxel/docetaxel and 133 (23.6%) in enfortumab vedotin were seen in the ureter (*p* = 0.574); and 108 (24.3%) in paclitaxel/docetaxel and 147 (26.1%) in enfortumab vedotin were seen in the renal pelvis (*p* = 0.531). A certain number of cases were duplicated, being diagnosed in each urothelial cancer location. The enfortumab vedotin group received more cystectomies than the paclitaxel/docetaxel group (96 (17.1%) in enfortumab vedotin, 49 (11.1%) in paclitaxel/docetaxel, *p* = 0.007). This study included the patients who received pembrolizumab. Due to the medical insurance policy, pembrolizumab was approved after patients received platinum based systemic chemotherapy. In this study cohort, all 98% of patients were confirmed to have received platinum-based treatment. And, the other 2% had a lack of data. Overall, 318 (71.8%) patients in paclitaxel/docetaxel and 517 (91.8%) in enfortumab vedotin received gemcitabine (*p* < 0.001); 217 (48.9%) in paclitaxel/docetaxel in 354 (62.9%) in enfortumab vedotin were received cisplatin (*p* < 0.001); 235 (53.0%) in paclitaxel/docetaxel and 234 (41.5%) in enfortumab vedotin received carboplatin (*p* < 0.001); and 24 (5.4%) in paclitaxel/docetaxel and 23 (4.3%) in enfortumab vedotin received methotrexate (*p* = 0.320) (Table 1).

In regard to prognosis, the enfortumab vedotin group showed a longer overall survival than the paclitaxel/docetaxel group (*p* = 0.013, HR: 0.71) (Figure 2). In multivariate analysis, enfortumab vedotin induction was an independent risk factor for overall survival (*p* = 0.013, HR: 0.70) (Table 2). There were no significant differences in cancer-specific survival (Appendix A and Appendix A).

## 4. Discussion

Enfortumab vedotin utilizes a unique mechanism of action that combines the specificity of an antibody–drug conjugate (ADC) with the potent cytotoxicity of a microtubule-disrupting agent [4,12]. The key components of this therapy are an anti-Nectin-4 monoclonal antibody and the cytotoxic agent monomethyl auristatin E (MMAE). Nectin-4 is a protein that is overexpressed in urothelial carcinoma cells, making it an ideal target for therapy [13]. The anti-Nectin-4 monoclonal antibody in enfortumab vedotin specifically binds to Nectin-4 on the cancer cells’ surface, allowing for the selective delivery of the cytotoxic agent to the tumor site. Once bound to Nectin-4, enfortumab vedotin is internalized by the cancer cells through receptor-mediated endocytosis. This process allows the ADC to enter the intracellular compartment, where it undergoes proteolytic cleavage, releasing MMAE [12]. MMAE is a potent inhibitor of microtubule polymerization, a crucial process for cell division and growth. By disrupting microtubules, MMAE interferes with the cancer cells’ ability to divide and proliferate, leading to cell cycle arrest and ultimately cell death. Furthermore, MMAE induces apoptosis, a programmed cell death mechanism, by triggering various cellular signals that activate the intrinsic apoptotic pathway [4]. This process involves the release of cytochrome c from the mitochondria, the activation of caspases, and the subsequent degradation of cellular components, ultimately resulting in cancer cell demise.

In this study, 563 patients were treated with enfortumab vedotin after pembrolizumab and their prognosis was compared with that of 463 patients treated with paclitaxel or docetaxel. This is the largest Japanese report to date. The EV-301 study included 301 patients in the enfortumab vedotin group and 307 patients in the chemotherapy group. Among them, the Japanese cohort included 36 in the enfortumab vedotin group and 50 in the chemotherapy (paclitaxel and docetaxel) group. Vinflunine was not used in the Japanese cohort due to it being a non-approved drug in Japan. One of the characteristics of the clinical trial is that only patients with a good performance status were included; thus, these results may not always represent the real-world results. And, because of differences in post-treatment due to the economic conditions in each country, it is difficult to evaluate the results in terms of OS for a specific country. Especially for Japan, all patients are covered by a medical insurance system and can receive the full scope of treatment options; thus, real-world data are needed. In this study, we were able to demonstrate the usefulness of enfortumab vedotin compared to systemic taxane chemotherapy after pembrolizumab in real-world Japanese patients.

In this study, the enfortumab vedotin group showed a longer OS than the paclitaxel/docetaxel group. A previous, large, phase Ⅲ, global randomized controlled study (EV-301) showed enfortumab vedotin as a life-prolonging agent compared to systemic chemotherapy [7]. Although the median survival for the enfortumab vedotin group was not reached due to the short observation period, the median survival of the paclitaxel/docetaxel group was similar to that of the chemotherapy group in the EV-301 study (in this study, paclitaxel/docetaxel cohort: 13.9 months, EV-301 study chemotherapy cohort: 9.97 months, and EV-301 study chemotherapy Japanese cohort: 10.6 months), and the HR was 0.74 in the EV-301 study and 0.70 in the present study [7,10]. These real-world data showed similar results compared to the EV-301 clinical trials. 

In this study, we found a significant difference in OS but not in CSS. This may be due to the small number of patients diagnosed as dying from cancer in the database analysis. In the current cohort, 251 (33.3%) cases were deaths, and 153 (18.0%) were cancer deaths. We suspect that the difference between the incidence of overall death and cancer death would be smaller in the situation of urothelial carcinoma patients receiving enfortumab vedotin or paclitaxel/docetaxel after pembrolizumab. It is highly likely that the insurance database recorded deaths as cancer deaths. Further studies with more detailed databases and longer observation periods would be warranted.

In the present study, the patients were compared with the chemotherapy group, the same as in the EV-301 study. In Japan, both paclitaxel and docetaxel are currently not approved by insurance for use in urothelial carcinoma, and so it is an off-label specification to use paclitaxel/docetaxel [14]. Therefore, the number of cases in the chemotherapy group was small. Before the insurance indication for enfortumab vedotin was approved in Japan, there was no established treatment option after pembrolizumab, and paclitaxel/docetaxel were used in the off-label setting.

This study did not include cases treated with avelumab. This study excluded these cases because the insurance approval of avelumab in Japan, based on the JAVELIN Bladder 100 study results, was in 2021; therefore, a sufficient observation period was not obtained, and the profile of urothelial carcinoma would be different from that of the pembrolizumab group because the JAVELIN Bladder 100 included only complete response, partial response, and stable disease cases after 4–6 courses of platinum-based chemotherapy [15]. The profile of urothelial carcinoma in the post-avelumab setting would be different from that of post-pembrolizumab settings.

This study has several limitations. The first is that this is a retrospective study and due to the insurance system database analysis, there is little patient information. On the other hand, there was no significant difference in the time from diagnosis to the start of treatment in each group, and there was no significant difference in age either, and so comparing paclitaxel/docetaxel and enfortumab vedotin for OS was not problematic. Second, this study included the patients enrolled in the Japanese medical insurance system and more than 99% were Asian Japanese. Thus, further study is needed for evaluating real-world data using worldwide data. Third, this study did not examine side effects; enfortumab vedotin is known to cause hematologic toxicity, hyperglycemia, skin rash, and other side effects. Although there was a difference in OS, the side effects should be examined in a different way in the future.

## 5. Conclusions

In conclusion, enfortumab vedotin prolonged the overall survival for Japanese advanced or metastatic urothelial carcinoma patients compared to paclitaxel or docetaxel treatment after pembrolizumab.

## Figures and Tables

**Figure 1 cancers-15-04227-f001:**
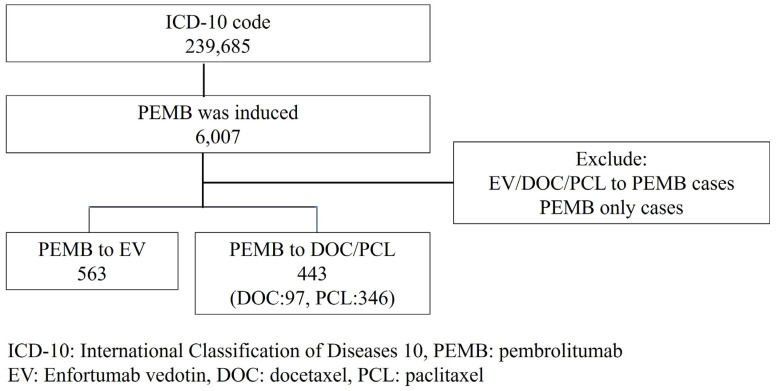
Patient selection.

**Figure 2 cancers-15-04227-f002:**
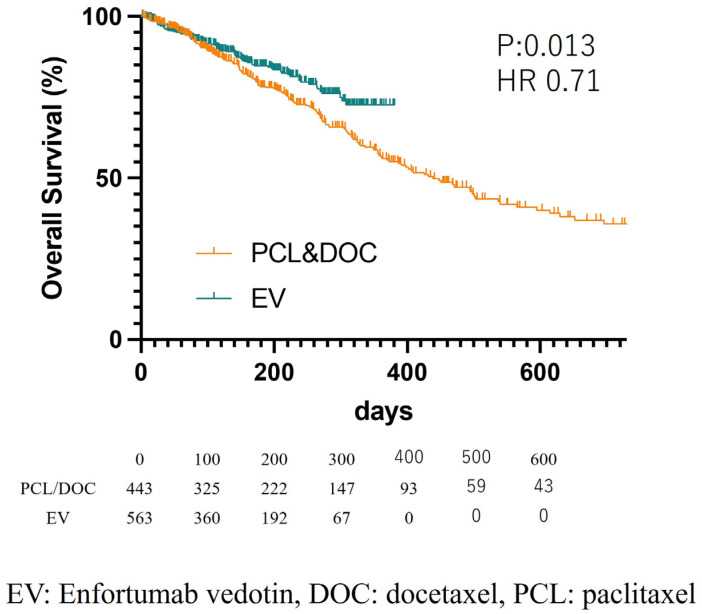
Kaplan–Meier Curve for overall survival: paclitaxel/docetaxel vs. enfortumab vedotin.

**Table 1 cancers-15-04227-t001:** Patients’ characteristics.

	Number (%), Median (Mean ± SD)			
	ALL	PCL/DOC	EV	*p*-Value (PCL/DOC vs. EV)
Number of patients	1006	443 (PCL346/DOC97)	563	
Male	784 (77.9%)	350 (79.0%)	434 (77.1%)	0.466
Age 70 or more (yrs.)	540 (53.7%)	235 (53.0%)	305 (54.2%)	0.772
Age at diagnosis (yrs.)	70 (69.2 ± 8.4)	70 (69.0 ± 8.5)	70 (69.4 ± 8.2%)	0.434
Age at each treatment (yrs.)	73 (71.8 ± 9.0)	73 (71.4 ± 9.9)	73 (72.2 ± 8.3)	0.187
Location				
Bladder	615 (61.1%)	279 (63.0%)	336 (59.7%)	0.287
Ureter	231 (23.0%)	98 (22.1%)	133 (23.6%)	0.574
Renal Pelvis	255 (25.3%)	108 (24.3%)	147 (26.1%)	0.531
Previous cystectomy	145 (14.4%)	49 (11.1%)	96 (17.1%)	0.007
Previous chemotherapy				
Gemcitabine	835 (83.0%)	318 (71.8%)	517 (91.8%)	<0.001
Cisplatin	571 (56.8%)	217 (48.9%)	354 (62.9%)	<0.001
Carboplatin	508 (50.5%)	235 (53.0%)	234 (41.5%)	<0.001
Methotrexate	48 (4.8%)	24 (5.4%)	23 (4.3%)	0.320
Pembrolizumab	1006 (100.0%)	443 (100.0%)	563 (100.0%)	1.000

PCL: paclitaxel, DOC: docetaxel, EV: enfortumab vedotin.

**Table 2 cancers-15-04227-t002:** Multivariable analyses for overall survival.

Variables	HR	95% CI		*p*-Value
		Lower	Upper	
Age 70 yrs. and over	0.87	0.68	1.11	0.266
Male	1.05	0.77	1.44	0.749
Bladder cancer	0.84	0.64	1.09	0.191
EV induction	0.70	0.53	0.93	0.013
Cystectomy	1.11	0.76	1.62	0.597

HR: hazard ratio, EV: enfortumab vedotin.

## Data Availability

The raw data to create tables and figures are available upon request.

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
