# Peer review of "Administration of Enfortumab Vedotin after Immune-Checkpoint Inhibitor and the Prognosis in Japanese Metastatic Urothelial Carcinoma: A Large Database Study on Enfortumab Vedotin in Metastatic Urothelial Carcinoma"

_cancers, 2023, doi:10.3390/cancers15174227_

Round 1

Reviewer 1 Report

The manuscript entitled “Administration of enfortumab vedotin after an immune check point inhibitor and the prognosis in Japanese metastatic urothelial carcinoma: A large database study” written by Kawahara et al, has well demonstrated beneficial effect of Enfortumab Vedotin (EV) on high grade urothelial carcinoma with improved overall survival in Japanese patients.

1)      Does the patient enrolled in the study had relapsed and/or refractory to platinum-based therapies?

2)      What significant adverse effect occurs in patients received EV and/or combinational therapy?

Limitations of the study:

1)      Powles et al., (NEJM, 2021) already reported OS effect of EV on large data set of metastatic urothelial carcinoma https://www.nejm.org/doi/10.1056, the present study lacks the novelty.

2)      The OS between EV vs regular first-line therapy was very modest.

3)      It would been better if author did a multi-centric study with large cohort, since the worldwide analysis made higher impact.

Overall, the study was well controlled and reported,

Author Response

Editorial office

Cancers

Re: cancers-2554335" Administration of enfortumab vedotin after immune-checkpoint inhibitor and the prognosis in Japanese metastatic urothelial carcinoma: A large database study".

Dear Editor,

Thank you for your letter concerning the abovementioned manuscript. We are pleased to note the favorable comments of the reviewers and have revised the manuscript. Our point-by-point revisions are described below.

We would like to thank the Editor and reviewers again for their helpful comments and hope that the revised manuscript is now acceptable for publication in Cancers.

Respectfully yours,

Takashi Kawahara, M.D., Ph.D.

Department of Urology and Renal Transplantation

Yokohama City University Medical Center

4-57 Urafune, Minami-ku, Yokohama, Kanagawa, 2320024, Japan

Phone: +81-45-787-2679

Fax: +81-45-786-5656

E-mail: [email protected] or [email protected]

RESPONSE TO REVIEWERS

Reviewer #1:

  • Does the patient enrolled in the study had relapsed and/or refractory to platinum-based therapies?

Response

We appreciate your comment. This study cohort included only Japanese patients and all patients were used Japanese medical insurance system. Currently, pembrolizumab is permitted after platinum-based treatment by Japanese insurance. Thus, all patients received platinum-based treatment basically. And in this cohort, all 98% of patients were confirmed to be received platinum-based treatment. And the other 2% were lack of data. We added this comment and limitation in the revised manuscript.

  • What significant adverse effect occurs in patients received EV and/or combinational therapy?

Response

Thank you for your comment. Unfortunately, this database did not include adverse effect data. We added limitation in the revised manuscript.

Limitations of the study:

  • Powles et al., (NEJM, 2021) already reported OS effect of EV on large data set of metastatic urothelial carcinoma https://www.nejm.org/doi/10.1056, the present study lacks the novelty.

Response

We appreciate your comment. We added the detailed of this data in the revised manuscript.

2)    The OS between EV vs regular first-line therapy was very modest.

Response

We appreciate your comment. We mistook to upload OS Kaplan-Miere curve figure. We inserted this results in the revised manuscript.

3)      It would been better if author did a multi-centric study with large cohort, since the worldwide analysis made higher impact.

Response

We appreciate your comment. As you pointing this out, worl-wide global study based on database is ideal. On the other hand, Asian-Japanese is quite different from global due to the sufficient medical insurance system and usually Asian-Japanese is god efficacy for systemic chemotherapy. We added these limitations in the revised manuscript.

Reviewer 2 Report

It is important research, with limitations that were mentioned by authors. It is retrospective study with limited patient's information focused on the Japanise cohort. However, it might be interesting for international readers. statistical approaches are adequate. In general, the manuscript can be published.

Author Response

Editorial office

Cancers

Re: cancers-2554335" Administration of enfortumab vedotin after immune-checkpoint inhibitor and the prognosis in Japanese metastatic urothelial carcinoma: A large database study".

Dear Editor,

Thank you for your letter concerning the abovementioned manuscript. We are pleased to note the favorable comments of the reviewers and have revised the manuscript. Our point-by-point revisions are described below.

We would like to thank the Editor and reviewers again for their helpful comments and hope that the revised manuscript is now acceptable for publication in Cancers.

Respectfully yours,

Takashi Kawahara, M.D., Ph.D.

Department of Urology and Renal Transplantation

Yokohama City University Medical Center

4-57 Urafune, Minami-ku, Yokohama, Kanagawa, 2320024, Japan

Phone: +81-45-787-2679

Fax: +81-45-786-5656

E-mail: [email protected] or [email protected]

RESPONSE TO REVIEWERS

Reviewer2

It is important research, with limitations that were mentioned by authors. It is retrospective study with limited patient's information focused on the Japanise cohort. However, it might be interesting for international readers. statistical approaches are adequate. In general, the manuscript can be published.

Response

We appreciate that you evaluated our study.

Reviewer 3 Report

The authors described the clinical data from the patients who received enfortumab vedotin and other chemotherapy (docetaxel or paclitaxel) after pembrolizumab. The authors included a larger cohort from Japan. A total of 6007 patients with urothelial cancer were treated with pembrolizumab. Among them, 619 patients received enfortumab vedotin, and 394 received docetaxel or paclitaxel. The authors found that enfortumab vedotin groups showed more prolonged survival than the docetaxel/paclitaxel groups. However, some concerns need to be addressed. 

1. It is confusing that in Table 1, the authors showed 443 patients with docetaxel/paclitaxel and 563 patients with enfortumab vedotin, but the number is not consistent with those authors described in Figure 1. 

2. The statistical method the authors used in Table 1 is unclear. 

3. The authors should present the KM-curve for the overall survival. 

4. The authors compared the patients treated with enfortumab vedotin and docetaxel/paclitaxel after pembrolizumab. Is there a difference between these patients and patients treated with pembrolizumab alone in overall and cancer-specific survival?

Author Response

Editorial office

Cancers

Re: cancers-2554335" Administration of enfortumab vedotin after immune-checkpoint inhibitor and the prognosis in Japanese metastatic urothelial carcinoma: A large database study".

Dear Editor,

Thank you for your letter concerning the abovementioned manuscript. We are pleased to note the favorable comments of the reviewers and have revised the manuscript. Our point-by-point revisions are described below.

We would like to thank the Editor and reviewers again for their helpful comments and hope that the revised manuscript is now acceptable for publication in Cancers.

Respectfully yours,

Takashi Kawahara, M.D., Ph.D.

Department of Urology and Renal Transplantation

Yokohama City University Medical Center

4-57 Urafune, Minami-ku, Yokohama, Kanagawa, 2320024, Japan

Phone: +81-45-787-2679

Fax: +81-45-786-5656

E-mail: [email protected] or [email protected]

RESPONSE TO REVIEWERS

Reviewer3

The authors described the clinical data from the patients who received enfortumab vedotin and other chemotherapy (docetaxel or paclitaxel) after pembrolizumab. The authors included a larger cohort from Japan. A total of 6007 patients with urothelial cancer were treated with pembrolizumab. Among them, 619 patients received enfortumab vedotin, and 394 received docetaxel or paclitaxel. The authors found that enfortumab vedotin groups showed more prolonged survival than the docetaxel/paclitaxel groups. However, some concerns need to be addressed.

  1. It is confusing that in Table 1, the authors showed 443 patients with docetaxel/paclitaxel and 563 patients with enfortumab vedotin, but the number is not consistent with those authors described in Figure 1.

Response

Thank you for your comment. We corrected the number in the revised manuscript.

  1. The statistical method the authors used in Table 1 is unclear.

Response

We appreciate your comment. We added the methods more easily to be understood for readers in the revised manuscript.

  1. The authors should present the KM-curve for the overall survival.

Response

We appreciate your comment. We mistook to upload the figure files. We re-uploaded the figure in the revised manuscript.

  1. The authors compared the patients treated with enfortumab vedotin and docetaxel/paclitaxel after pembrolizumab. Is there a difference between these patients and patients treated with pembrolizumab alone in overall and cancer-specific survival?

Response

We appreciate your comment. We also examined the cohort who received pembrolizumab only. As shown in this figure, there were big difference between pembrolizumab only group vs pembrolizumab-EV301 group. We speculate that the pembrolizumab only group included the patients who received radical cystectomy or nephroureterectomy after pembrolizumab treatment. Thus, we set the cohort EV301 vs DOC/PCL after pembrolizumab treatment. This comparison is same as EV301 study. We added these limitation in the revised manuscript.

Round 2

Reviewer 3 Report

Accept in present form.